# Hepatoprotective Effect of Alpinetin on Thioacetamide-Induced Liver Fibrosis in Sprague Dawley Rat

Suhayla Hamad Shareef [1,*], Ameena S. M. Juma [2], Derin N. F. Agha [3], Abdullah R. Alzahrani [4], Ibrahim Abdel Aziz Ibrahim [4] and Mahmood Ameen Abdulla [2]

1 Department of Biology, College of Education, Salahaddin University-Erbil, Erbil 44001, Iraq
2 Department of Medical Microbiology, College of Science, Cihan University-Erbil, Erbil 44001, Iraq
3 Department of Pharmacology, Erbil Medical Technical Institute, Erbil Polytechnic University, Erbil 44001, Iraq
4 Department of Pharmacology and Toxicology, Faculty of Medicine, Umm Al-Qura University, Makkah 77207, Saudi Arabia
* Correspondence: suhayla.shareef@su.edu.krd; Tel.: +964-07728414580

**Abstract:** Alpinetin is an original medicinal plant flavonoid derived from *Alpinia katsumadai* and has several biological activities. The current research aimed to evaluate the hepatoprotective effects of Alpinetin against thioacetamide (TAA)-induced liver cirrhosis in rats. Five groups of rats were utilized in this study. Hepatic injury was measured macroscopically and microscopically for entire groups. The rats' body weight was significantly lower in the TAA control group, likened to rats fed with Silymarin or Alpinetin groups, while liver weight was significantly greater in the TAA control group when equated to rats nourished with Alpinetin groups. A histopathological investigation of hepatic tissues displayed that TAA remarkably induced hepatocyte necrosis and gristly connective tissue propagation in the TAA control group. Alpinetin implicitly decreased the influence of TAA toxicity and diminished fibrosis of liver tissues. The TAA control group presented an increase in liver enzymes (ALP, ALT, and AST) and a decrease in total protein and albumin. Rats who were fed Alpinetin had significantly lower hepatic enzyme activity as well as augmented total protein and albumin, yet they were close to the normal range. Superoxide dismutase (SOD) and Catalase (CAT) enzymes in hepatic homogenate were significantly reduced, and malondialdehyde (MDA) was meaningfully elevated in the TAA control group, while rats fed with Alpinetin had significantly increased SOD and CAT achievement and depressed MDA level. Alpinetin-gavaged groups had reduced levels of Tumor necrosis factor-alpha (TNF-$\alpha$) and Interleukin-6 (IL-6), significantly down-regulated Proliferating cell nuclear antigen (PCNA), Alpha-smooth muscle ($\alpha$-SMA), and reduced hepatic stellate cell activity. However, the TAA control group significantly up-regulated PCNA and $\alpha$-SMA and increased the activity of hepatic stellate cells. Alpinetin was nontoxic and could improve defensive mechanisms against hepatic tissue injury. Acute toxicity tests discovered no evidence of any toxic signs or dead rats, which highlights the safety of Alpinetin. Consequently, the investigation´s outcomes revealed that the hepatoprotective effects of Alpinetin in TAA-induced hepatic impairment might be due to reduced TAA toxicity, increased protein and albumin, increased SOD and CAT levels, reduced MDA levels, and modulation of inflammatory cytokines and their anti-oxidant activities, and suppressed PCNA and $\alpha$-SMA.

**Keywords:** liver fibrosis; thioacetamide; Alpinetin; hepatoprotective; Silymarin; histopathology; $\alpha$-SMA

## 1. Introduction

The liver is an important organ responsible for the removal of tissue breakdown in the body. Chemical changes from diverse materials enhance the formation of reactive oxygen species [1]. Released radicals form hepatotoxins, for instance, thioacetamide. The role of

TAA hepatotoxic mediators as contributors to liver damage and long-lasting management of fibrosis of the liver has been reported by numerous researchers [2–6].

TAA is hepatotoxic; a single dose is accepted to be able to yield centrilobular hepatocyte necrosis, and continuing administration led to liver fibrosis in laboratory animals [7,8]. High dosages lead to liver cirrhosis and hepatocarcinomatous [9]. Numerous researchers have used diverse approaches in the TAA management of laboratory rats for creating cirrhosis [4,10–12]. Aromatic plants and therapeutic basic medication materials facilitate their use as remedies against many diseases, including liver cirrhosis. In the methodical works, huge quantities of remedial plants with hepatoprotective effects have been described [4,10,13–15].

Alpinetin (7-hydroxy-5-methoxy flavanone; C16H14O4) is an ordinary flavonoid primarily resulting from the roots or seeds of *Alpinia katsumadai Hayata*, which is known to display antibacterial, anti-inflammatory, and additional significant beneficial effects [16]. Alpinetin, the usual flavonoid, is the main lively constituent of the traditional remedial herb *Alpinia katsumadai Hayata* and has been used therapeutically since prehistoric times [17]. Alpinetin has been described to have anticancer properties by several mechanisms, including apoptosis induction, cell cycle stop, and proliferative clampdown in numerous kinds of cancer, for example, breast cancer, lung cancer, colon cancer, and liver cancer [18]. Furthermore, regarding anticancer activity, Alpinetin has revealed substantial anti-inflammatory actions. Alpinetin has been reported to prevent NF-κB activation and reduction oxidative stress from overpowering the relief of pro-inflammatory cytokines, for instance, TNF-α and IL-1 [19].

Alpinetin has protective effects on DSS-induced colitis and may be a promising therapeutic reagent for colitis treatment [17]. Alpinetin might have anticancer properties on human ovarian cancer by preventing the STAT3 signaling pathway [18]. Alpinetin is stated to inhibit the propagation of human tumor cells, comprising lung cancer and gastroenteric cancer cells [20], representing the potential anticancer properties of this compound. Alpinetin-enhanced Nrf2-mediated redox homeostasis, which subsequently repressed macrophage penetration and atherosclerosis, signifying a valuable compound for handling atherosclerosis [21].

Alpinetin is the original flavonoid constituent of numerous edible and medical plants and can provide an extensive range of biological and pharmacological actions, including antibacterial, anti-hemostatic, anti-oxidative, anti-hepatotoxic, stomachic, immunosuppressive, and anti-inflammatory effects [22]. Alpinetin is also used for oncological treatment [22]. Alpinetin and its diverse metabolites, for example, glucuronide conjugates, prototype, and phenolic acids, are responsible for its several pharmacological functions. These functions comprise antibacterial, anti-hemostatic, anti-oxidant, anti-hepatotoxic, immunosuppressive, anti-inflammatory, and stomachic effects [16].

Alpinetin is used in traditional medications; it is consumed as part of the human diet and used in the progress of new medications [23]. Alpinetin has condensed lipopolysaccharide (LPS) encouraged lung [24], kidney damage [25], and mastitis [26] in dextran sulfate sodium-induced colitis in mice [27]. Moreover, Alpinetin showed a robust possibility to prevent LPS-induced inflammatory response in countless in vitro and in vivo studies [28].

In the animal model, Alpinetin condensed the alveolus pulmonic cells' damage by overturning expressions of IL-6, α-SMA, TNF-α, and transmuting growth factor-β1 (TGF-β1) related to a huge failure of caspase-3 and caspase-9 and upregulation IL-10 [29].

Up to now, there is no information accessible on the hepatoprotective activity of Alpinetin. The existing study is conducted to evaluate possible hepatoprotective possessions of Alpinetin in TAA-induced liver cirrhosis in rats.

## 2. Materials and Methodologies

### 2.1. Conscience Declaration

The existing study was authorized through the ethics team examination, College Science, Cihan University-Erbil, Ethic number ERB, 125, 17/08/2022. Altogether rats (Sprague

Dawley), during the period of study, received human care in agreement with values usual onward via the "Executive Conservation Usage investigation test center" prearranged through the Nationwide School of Knowledge and delivered via the Nationwide Institution of Healthiness.

## 2.2. Preparation of Alpinetin

Alpinetin was obtained from Sigma–Aldrich Chemical Co. (St. Louis, MO, USA). Alpinetin was dissolved in 10% of Tween 20, then Sprague Dawley rats administrated dosages were 30 and 60 mg/kg (5 mL/kg) [30].

## 2.3. Preparation of Thioacetamide

Thioacetamide was purchased from Sigma–Aldrich (St. Louis, MO, USA), and liquefied in 10% Tween 20. Two hundred mg/kg were inoculated i.p. to the Sprague Dawley rats three times weekly for two months. TAA alters the biochemical and morphologic features of the liver, analogous to human liver cirrhosis [31].

## 2.4. Preparation of Silymarin

Silymarin was obtained from the International Laboratory, Sigma-Aldrich, St. Louis, MO, USA, positive control medication and dissolved in 10% Tween 20, given by mouth dosages of 50 mg/kg [31].

## 2.5. Acute Toxicity Test and Experimental Rats

Thirty-six (18 males and 18 females) pathogen-free rats (7–8 weeks old, considering between 170 and 190 g) were obtained from Animal House, Cihan University-Erbil. Rats were given normal rat pellets and tap water ad libitum and located in separate cages with wide-mesh wire bases to prevent coprophagia. Rats were reserved in cages for one week for adaptation. An acute toxicity study was used to determine the harmless dosage of Alpinetin. Rats were assigned three groups: 10% Tween 20, 30 mg/kg, and 300 mg/kg Alpinetin. Before supplement and treatment, overnight fasting was applied to altogether rat groups. Nourishment was uninvolved for about 3–4 h. The animals were observed for 24–48 h after the administration of the Alpinetin, the rats were checked for starting systematic toxicological symbols. The mortality degree was calculated for two weeks. Animals were injected with overdose doses of xylazine and ketamine anesthesia on the 15th day. Blood samples were collected through intracranial puncture; serum was separated, and samples were taken for biochemistry parameters examination. Histology and serum biochemistry constraints resolute succeeding typical procedures [31].

## 2.6. Laboratory Investigational Sprague Dawley Rats

Matured healthy pathogen-free Sprague Dawley rats (7–8 weeks old, 170–190 g) acquired from Investigational Animal Unit, Cihan University-Erbil. Rats were stored separately in cages with wide-network chain bases. Experimental rats were upheld on normal pellet food and tap water and adapted to distinctive workroom conditions for one week before the experimentation [32].

## 2.7. Experimental Procedures

In this study, all experimental procedures were accepted via the ethical board for rats' examination. Thirty adult mature male rats were arbitrarily separated into five groups, six animals separately as shadows:

Group 1 (normal group) rats nourished orally 10% Tween 20 (5 mL/kg) every day and inoculated intraperitoneal (i.p) 10% Tween 20 (5 mL/kg) three times weekly for two months.
Group 2 (TAA group) rats fed by mouth 10% Tween 20 (5 mL/kg) injected i.p TAA (200 mg/kg) three times weekly for two months.
Group 3 (Silymarin group) rats were administered orally Silymarin (50 mg/kg) daily injected i.p TAA (200 mg/ kg) three times every week for two months.

Groups 4 and 5 (Alpinetin gavaged groups) rats nourished by mouth Alpinetin 30 mg/kg and 60 mg/kg daily, correspondingly. Rats have inserted i.p TAA (200 mg/kg) three times weekly for two months [33].

The body weight of laboratory rat groups was taken weekly [34]. Two months later, whole rats were starved under euthanized anesthesia using Ketamine (30 mg/kg, 100 mg/mL) and Xylazine (3 mg/kg, 100 mg/mL). Blood was collected from heart perforation, located in gel-activated plain tubes, spun, and transferred to Laboratory Rizgary Hospital for biochemical analysis [34].

## 2.8. Macroscopically Examination of Liver

The livers were separated, rinsed with cold phosphate-buffered saline, dried with filter papers, and inspected for weight and gross pathological alterations [35].

## 2.9. Histopathological Investigation of Liver

Hepatic tissue was fixed with 10% phosphate-buffered formalin complete fixation and implanted in wax using tissue processing apparatus (Leica, Germany). Sections 3–5 μm thick were prepared and stained with Hematoxylin and Eosin stain [34] and a special stain (Masson Trichrome) [36] for histological valuation.

## 2.10. Immunohistochemistry

Liver tissue slices were heated in an oven at 60 °C for 30 min, de-parasitized in xylene, and rehydrated in categorized alcohol. The antigen recovery method exhibited 10 mM sodium citrate buffer boiled in a microwave. The staining technique was shown according to the manufacturer's instructions. Concisely, 0.03% hydrogen peroxide sodium azide was utilized for hunk endogenous peroxidase for 5 min; then, slices were washed away. Afterward, washed slices were incubated with biotinylated primary antibodies against (PCNA; 1:200) for 15 min, and rabbit anti-rat $\alpha$-smooth muscle for actin antibody (1:100) for 30 min. Tissue slices were mildly rinsed in distilled water for 3 min and reserved in a buffer bath in the moist compartment. Streptavidin HRP added slices were incubated for 15 min and washed in twice-distilled water for 3 min. Biopsy slices incubated diaminobenzidine (DAB) substrate Chromogen for 5 min, rinsed counterstained hematoxylin for 5 s. Lastly, slices were immersed 10 times in weak ammonia (0.037 M/L), washed with distilled water for 2–5 min, and mounted rising medium. Below the compound microscope, positive antigens stain brown against blue hematoxylin background. Propagation index PCNA-stain hepatic slices were measured by counting the percentage of labeled cells per 1000 liver cells, several number mitotic cells articulated mitotic index [10].

## 2.11. Endogenous Anti-Oxidant Enzymes Estimation

Liver tissue specimens were taken from identical sites of the left lobe of the whole liver. One gram of liver parenchyma was immersed in 10 mL (10%) PBS at pH 7.2 before being homogenized by a homogenizer and spun at 5000 rpm for 15 min at $-4$ °C. The top clear part was secluded and reserved in a $-80$ °C freezer, and analyses were done by the rendering guide of the producer. Commercial kits (Cayman Chemical Co., Ann Arbor, MI, USA) were utilized to evaluate MDA, CAT, and SOD [6].

## 2.12. Biochemistry of Liver Functions

Blood was collected from each rat, and serum was separated to measure liver function enzymes, including ALT, AST, ALP, bilirubin, albumin, and protein [36].

## 2.13. Evaluation of Inflammatory Cytokines (TNF-α and IL-6)

Evaluation of TNF-$\alpha$ and IL-6 was achieved employing the commercial ELISA kit BioSource, following the procedure written in the kit instructions in Rat TNF-$\alpha$ ELISA Kit (MBS267737) and Rat IL-6 ELISA Kit (MBS355410).

### 2.14. Statistical Analysis

The data of this study were analyzed using a one-way analysis of variance (ANOVA) shadowed via Bonferroni's post hoc examination (SPSS version 22, SPSS Inc., Chicago, IL, USA). Outcomes selected as mean ± S.E.M.; prospect value $p < 0.05$ measured as significance.

## 3. Results

### 3.1. Acute Toxicity Test

Acute destructiveness assessment displays thirty-six (18 males and 18 females) no representation of noxiousness. No histopathology marks the liver besides renal harmfulness. Furthermore, blood biochemical investigation seemed normal. The toxicity test of rats fed on Alpinetin exhibited no death or lethal signs throughout the experimental study. No abnormal physiological or behavior differences were found in dose 30 besides 300 mg/kg subsequent Alpinetin gavage (Tables 1–4). Histopathology examination in addition to biochemistry assessment of the liver and kidney (Figure 1, Tables 1–4). Afterward, the SD rats did not display any important symbols of destruction.

**Table 1.** Effects of Alpinetin on hepatic biochemistry parameters in the ($n = 18$) female rats.

| Dose | Albumin (g/L) | Globulin (g/L) | TB (μmol/L) | ALP (IU/L) | ALT (IU/L) | AST (IU/L) |
|---|---|---|---|---|---|---|
| Vehicle | 13.29 ± 0.04 | 66.12 ± 0.22 | 1 | 95.90 ± 0.24 | 39.37 ± 0.18 | 196.0 ± 0.51 |
| | | (10% Tween 20) | | | | |
| Alpinetin 30 mg/kg | 13.50 ± 0.05 | 66.40 ± 0.19 | 1 | 97.98 ± 0.34 | 40.23 ± 0.35 | 197.4 ± 0.50 |
| Alpinetin 300 mg/kg | 13.19 ± 0.16 | 65.70 ± 0.32 | 1 | 96.65 ± 0.96 | 39.11 ± 0.37 | 196.2 ± 0.45 |

Data stated mean ± S.E.M. No substantial variations among groups, and significant value at $p < 0.05$. Abbreviations: ALP, alkaline phosphatase; ALT, alanine aminotransferase; AST, aspartate aminotransferase; TB, total bilirubin.

**Table 2.** Effects of Alpinetin on liver biochemical parameters in the ($n = 18$) male rats.

| Dose | Albumin (g/L) | Globulin (g/L) | TB (μmol/L) | ALP (IU/L) | ALT (IU/L) | AST (IU/L) |
|---|---|---|---|---|---|---|
| Vehicle | 11.55 ± 0.21 | 57.63 ± 1.70 | 1 | 199.2 ± 0.51 | 56.54 ± 0.36 | 214.2 ± 1.45 |
| | | (10% Tween 20) | | | | |
| Alpinetin 30 mg/kg | 11.47 ± 0.06 | 57.07 ± 0.23 | 1 | 197.1 ± 0.65 | 57.18 ± 0.28 | 214.3 ± 1.01 |
| Alpinetin 300 mg/kg | 11.96 ± 0.28 | 56.37 ± 0.19 | 1 | 197.5 ± 0.59 | 57.11 ± 0.41 | 210.3 ± 1.32 |

Values are stated as mean ± S.E.M. No significant variations between groups, the significant value of $p < 0.05$. Abbreviations: ALP, alkaline phosphatase; ALT, alanine aminotransferase; AST, aspartate aminotransferase; TB, total bilirubin.

**Table 3.** Influence of Alpinetin on kidney biochemistry parameters in the ($n = 18$) female rats.

| Dose | Sodium (mmol/L) | Potassium (mmol/L) | Chloride (mmol/L) | Urea (mmol/L) | Creatinine (μmol/L) |
|---|---|---|---|---|---|
| Vehicle (10% Tween 20) | 146.0 ± 0.25 | 4.73 ± 0.02 | 106.1 ± 0.27 | 7.42 ± 0.15 | 36.10 ± 0.27 |
| Alpinetin 30 mg/kg | 146.9 ± 0.17 | 4.79 ± 0.01 | 106.4 ± 0.19 | 8.29 ± 0.18 | 36.84 ± 0.35 |
| Alpinetin 300 mg/kg | 146.5 ± 0.34 | 4.79 ± 0.01 | 106.9 ± 0.23 | 7.45 ± 0.35 | 36.60 ± 0.16 |

Standards were specified as mean ± S.E.M. No substantial variations among groups were significant, $p < 0.05$.

**Table 4.** Influence of Alpinetin on kidney biochemistry parameters in the ($n = 18$) male rats.

| Dose | Sodium (mmol/L) | Potassium (mmol/L) | Chloride (mmol/L) | Urea (mmol/L) | Creatinine (μmol/L) |
|---|---|---|---|---|---|
| Vehicle (10% Tween 20) | 145.8 ± 0.22 | 5.48 ± 0.16 | 104.9 ± 0.40 | 5.58 ± 0.18 | 30.85 ± 0.33 |
| Alpinetin 30 mg/kg | 145.3 ± 0.07 | 5.82 ± 0.01 | 104.9 ± 17.0 | 5.62 ± 0.10 | 31.18 ± 0.25 |
| Alpinetin 300 mg/kg | 145.3 ± 0.06 | 5.80 ± 0.02 | 104.4 ± 0.09 | 5.45 ± 0.11 | 30.93 ± 0.24 |

Standards were specified as mean ± S.E.M. No substantial variations among groups were significant, $p < 0.05$.

Liver Kidney

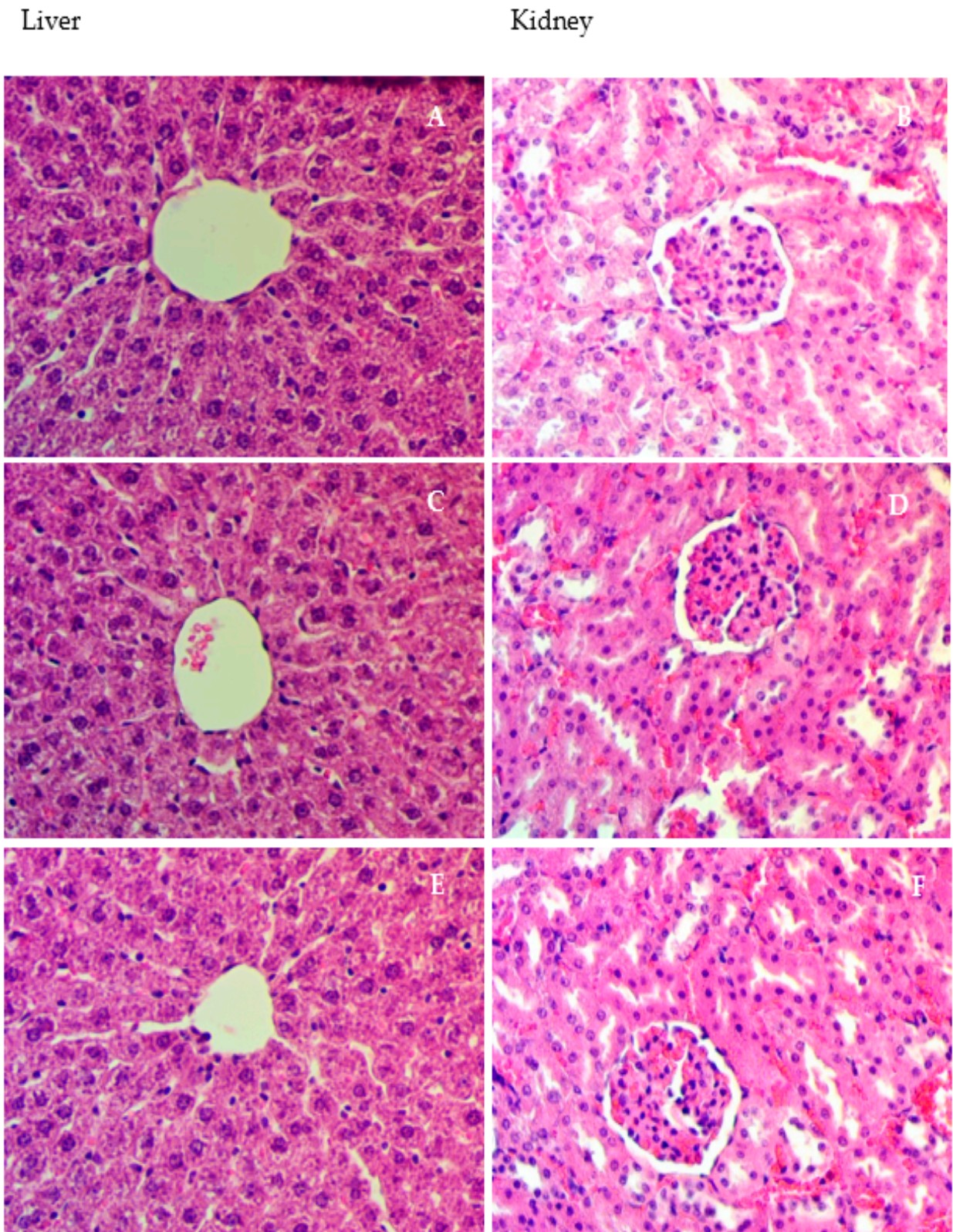

**Figure 1.** Histologic slices of hepatic besides kidney for acute toxicity experimentation. Animals fed 5 mL/kg vehicle (10% Tween 20) (**A**,**B**). Rats fed on 30 mg/kg (5 mL/kg) Alpinetin (**C**,**D**). Rats fed on 300 mg/kg of Alpinetin (**E**,**F**). No substantial variations in liver and kidney constructions between the experimental and control groups were seen (hematoxylin and eosin stain 40×).

### 3.2. Body and Liver Masses

The mass of rats in the TAA control group was meaningfully inferior, in addition to a rise in liver mass in comparison to the normal group. Rats gavaged Silymarin or Alpinetin implicitly exhibited enhanced body weight and a drop in their liver bulks (Table 5).

**Table 5.** Influence of Alpinetin on body mass and liver heaviness in TAA-induced liver cirrhosis experimental rats.

| Groups | Body Weight (gm) | Liver Weight (gm) | Liver Index (%) LW/BW% |
|---|---|---|---|
| Normal Control | 330 ± 3.72 [b] | 10.15 ± 0.08 [b] | 3.078 ± 0.01 [b] |
| TAA Control | 168 ± 1.59 [a] | 12.87 ± 0.03 [a] | 7.66 ± 0.06 [a] |
| Silymarin (50 mg/kg) + TAA | 306.16 ± 4.17 [b] | 10.32 ± 0.06 [b] | 3.37 ± 0.06 [b] |
| Alpinetin (30 mg/kg) + TAA | 231 ± 1.88 [b] | 10.45 ± 0.04 [b] | 4.52 ± 0.02 [b] |
| Alpinetin (60 mg/kg) + TAA | 247.66 ± 1.54 [b] | 10.22 ± 0.03 [b] | 4.127 ± 0.03 [b] |

The value was represented by mean ± S.E.M ($n = 6$). According to Bonferroni's post hoc test at a $p < 0.05$ significant level, the values of different superscripts within the same column are significantly different. Superscript a and b are used for describing the difference between groups, if (a and b) present in the same column it means a significant difference between groups.

### 3.3. Effect of Alpinetin on Anti-Oxidant Enzymes

SOD and CAT enzyme activities in hepatic homogenate were significantly reduced as hepatotoxic groups in relation to the normal group. Rats orally given Silymarin or Alpinetin presented significantly increased levels of SOD and CAT compared to the hepatotoxic group (Table 6). Oxidative stress in liver parenchyma was calculated by determining the lipid peroxidation (MDA) value in hepatic tissue homogenates. MDA levels were expressively increased in TAA control animals in contrast normal group. Rats that received oral Silymarin or Alpinetin perceptively displayed a lesser MDA likened TAA control group, reinstating level analogous to the normal rats (Table 6).

**Table 6.** Results of Alpinetin on SOD, CAT, and MDA content in liver homogenate in the TAA-induced liver fibrosis in rats.

| Groups | SOD U/mg Protein | CAT U/mg Protein | MDA U/mg Protein |
|---|---|---|---|
| Normal Control | 15.87 ± 0.60 [b] | 36.84 ± 0.8 [b] | 1.18 ± 0.01 [b] |
| TAA Control | 8.35 ± 0.24 [a] | 16.95 ± 0.52 [a] | 4.62 ± 0.13 [a b] |
| Silymarin (50 mg/kg) +TAA | 12.21 ± 0.26 [b] | 28.07 ± 0.57 [b] | 1.54 ± 0.09 [b] |
| Alpinetin (30 mg/kg) + TAA | 13.92 ± 0.05 [b] | 32.78 ± 0.90 [b] | 2.16 ± 0.02 [b] |
| Alpinetin (60 mg/kg) + TAA | 13.09 ± 0.30 [b] | 30.81 ± 0.78 [b] | 1.87 ± 0.03 [b] |

The value was represented by mean ± S.E.M ($n = 6$). According to Bonferroni's post hoc test at a $p < 0.05$ significant level, the values of different superscripts within the same column are significantly different. Superscripts a and b are used for describing the difference between groups, if (a and b) present in the same column it means a significant difference between groups.

### 3.4. Influence of Alpinetin on Biochemical Parameters

TAA-induced impairment in hepatic tissues and the grade of injury were measured by substantial serum amounts of specific hepatic enzymes, for instance, ALP, ALT, AST, and bilirubin. Liver enzymes appeared provocatively raised in the TAA control group owing to show liver enzymes as a consequence of tissue damage. Rats fed orally with Silymarin or Alpinetin exhibited significantly lesser enzyme activity and bilirubin (Table 7).

**Table 7.** Effects of Alpinetin in biochemistry parameters on the TAA-induced liver cirrhosis in rats.

| Groups | ALP (IU/L) | ALT (IU/L) | AST (IU/L) | T.Bilirubin (uM) | Protein (g/L) | Albumin (g/L) |
|---|---|---|---|---|---|---|
| Normal Control | 98.01 ± 1.76 [b] | 60.92 ± 1.53 [b] | 166.41 ± 4.95 [b] | 3.55 ± 0.05 [b] | 65.34 ± 1.98 [b] | 12.50 ± 0.54 [b] |
| TAA Control | 231.32 ± 1.70 [a] | 203.16 ± 3.44 [a] | 309.42 ± 3.28 [a] | 8.84 ± 0.07 [a] | 50.15 ± 2.14 [a] | 7.32 ± 0.154 [a] |
| Silymarin (50 mg/kg) + TAA | 126.73 ± 3.96 [b] | 74.67 ± 1.90 [b] | 184.12 ± 4.75 [b] | 5.27 ± 0.10 [b] | 63.38 ± 1.65 [b] | 12.351 ± 0.13 [b] |
| Alpinetin (30 mg/kg) + TAA | 150.98 ± 3.49 [b] | 85.30 ± 2.20 [b] | 210.50 ± 3.19 [b] | 6.05 ± 0.05 [b] | 59.28 ± 1.76 [b] | 11.72 ± 0.31 [b] |
| Alpinetin (60 mg/kg) + TAA | 142.13 ± 3.64 [b] | 77.99 ± 2.31 [b] | 193.51 ± 2.90 [b] | 5.80 ± 0.30 [b] | 62.15 ± 1.74 [b] | 12.28 ± 10.17 [b] |

The value was represented by mean ± S.E.M ($n = 6$). According to Bonferroni's post hoc test at a $p < 0.05$ significant level, the values of different superscripts within the same column are significantly different. Superscripts a and b are used for describing the difference between groups, if (a and b) are present in the same column it means a significant difference between groups.

The protein, besides albumin values, perceptively substandard hepatotoxic group associated rats fed with Silymarin or Alpinetin groups, representing acute hepatocellular injury. Silymarin or Alpinetin-fed rats reinstated the protein and albumin closer to their normal ranges. Therefore, Silymarin or Alpinetin responded to the toxic consequence of TAA by returning to standard liver functions (Table 7).

### 3.5. Morphology of Liver

The liver of the usual control rats looked uniform with smooth surfaces. However, the liver of the TAA-controlled group formed liver cirrhosis. The liver seems coarse, irregular external surface nodular, with even micro-nodules and macro-nodules superficial (Figure 2). Animals fed with Silymarin or Alpinetin strangely improved the retrieval of liver construction from the harm induced via TAA and safe the liver from extra worsening.

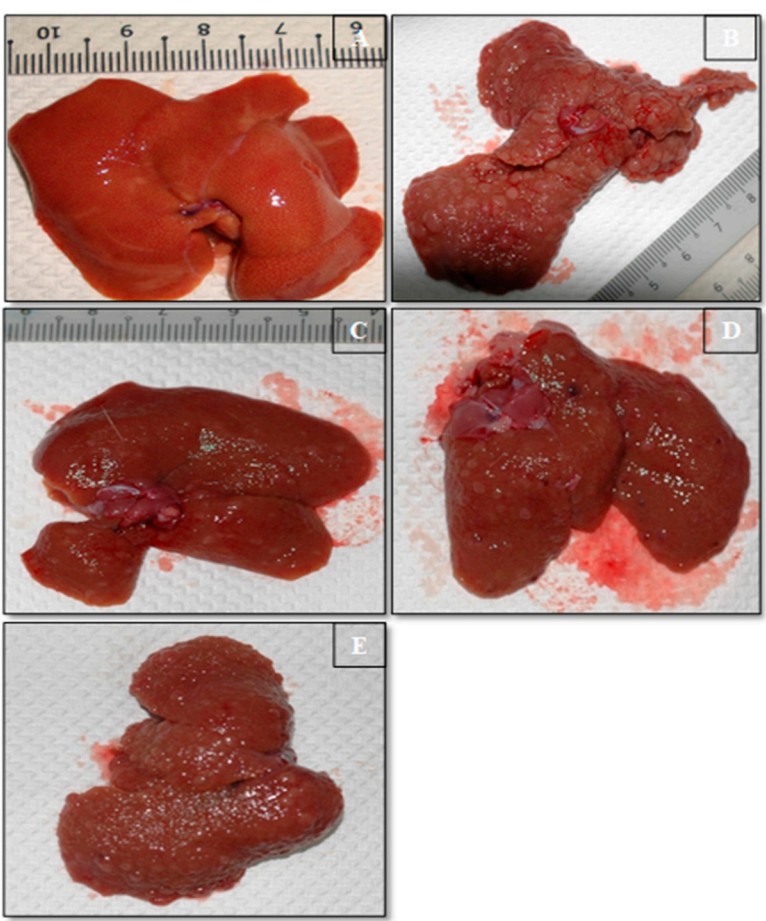

**Figure 2.** Gross Liver Morphology. (**A**) The normal control group showed uniformity, even external

superficiality. (**B**) TAA control group displayed coarse nodular external, with even dispersal of micronodules and macronodules. (**C**) TAA + Silymarin group demonstration was even flat outward. (**D**) TAA + 30 mg/kg Alpinetin liver showed close, even superficial, and limited micronodules. (**E**) TAA + 60 mg/kg Alpinetin liver showed typical even superficial, and closely regular structural form and appearance.

### 3.6. Microscopically Investigation of Liver

The control group presented regular liver with hepatic cords, sinusoidal spaces, and a principal vein. Liver cells have integral well-maintained cytoplasm and prominent nuclei. Livers were free from pathological aberration, and their slices were observed with even cellular construction. The histological structures of liver parenchyma tissues were integral (Figures 3 and 4).

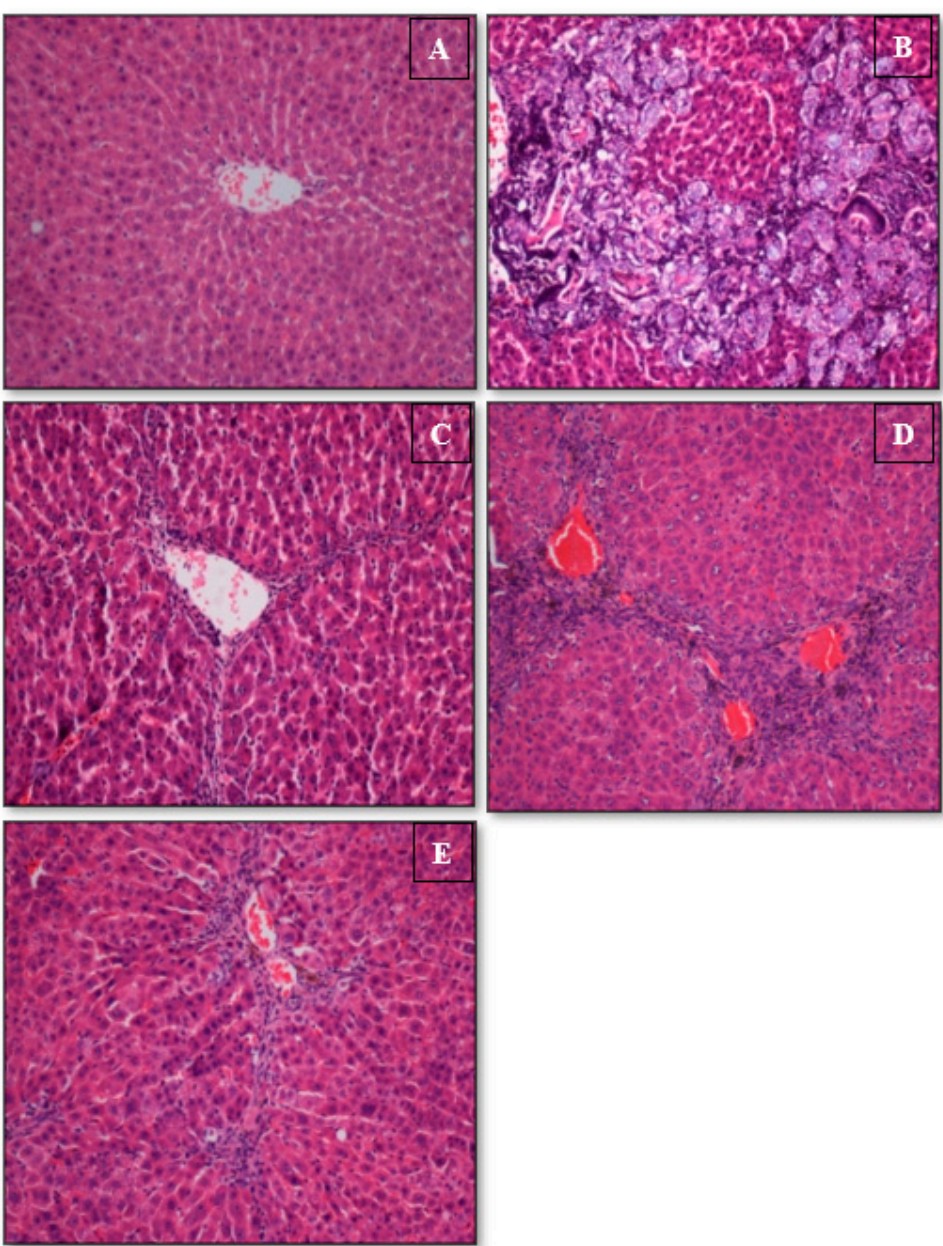

**Figure 3.** Histopathology of liver sections. (**A**) The normal group presented the normal histological

structure and style. (**B**) TAA group exhibited physical injury, asymmetrical renewing lobules with a condensed fibrotic septum, proliferation of bile ducts, and existence of inflammation. (**C**) TAA + Silymarin liver with minor inflammation and fibrosis. (**D**) TAA + 30 mg/kg Alpinetin group exposed moderately well-maintained hepatic cells, a small range of necrosis, and a thin fibrotic septum. (**E**) TAA+ 60 mg/kg Alpinetin group presented partly conserved hepatocytes and minor zones of insignificant necrosis (H&E stain 20×).

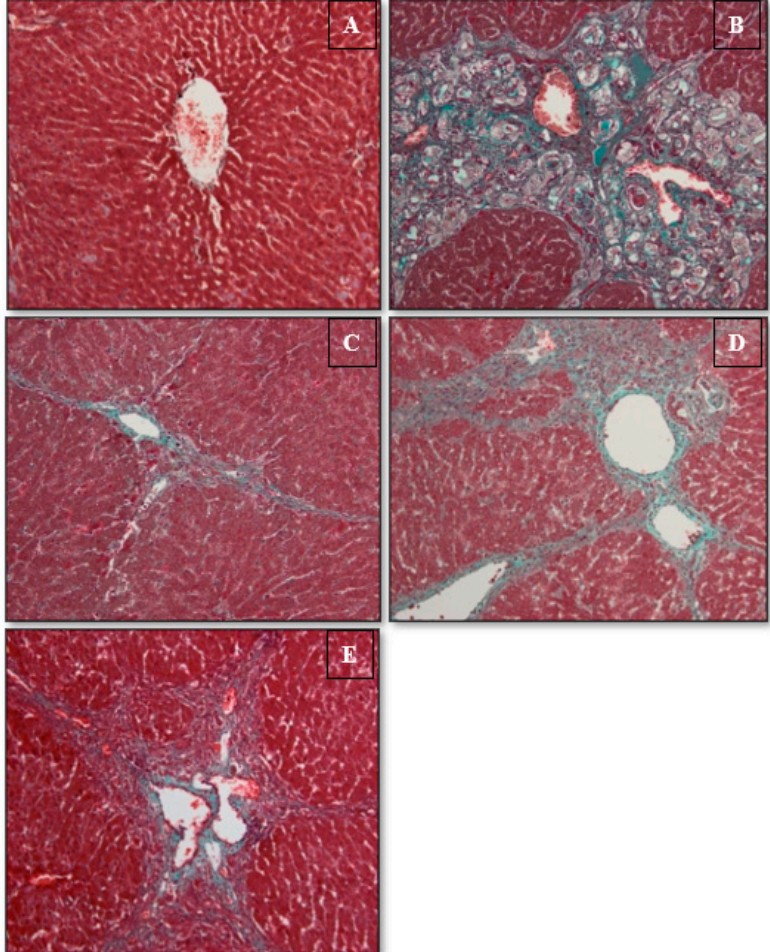

**Figure 4.** Histopathology of hepatic slices. (**A**) The usual group exhibited standard structure. (**B**) TAA control group displays propagation of bile ducts, thick rubbery septum, besides collagen threads. (**C**) TAA + Silymarin group exhibits slight leathery septa and collagen filaments. (**D**) TAA + 30 mg/kg Alpinetin group demonstrates reasonable fibrous septum and uneven restoring nodules. (**E**) TAA + 60 mg/kg Alpinetin group displayed a slightly tough septum besides collagen threads (Masson's trichrome stains 20×).

In the TAA control group, histology proved liver damage was demonstrated through the presence of inflammation besides necrotic tissue debris in corresponding hepatic sections. The construction of hepatic tissue is changed by fibrous septa arranged collagen connection among hepatic triangles outlining minor and big re-forming nodules of necrotic tissue. The nodules bounded via dense fibrous connective tissue separated the liver into virtual lobules. The representations of cytoplasmic vacuolation, bile vessel propagation, hepatic cell deterioration, necrosis, a gathering of inflammatory cells, extended portal areas, and collagen confession were observed.

The Silymarin-fed group revealed significantly minor pathological changes as correlated to a widespread liver injury initiated in the hepatotoxic group. Silymarin prevented inflammation, cellular permeation, liver cell death, then fibrous connection material pro-

duction displayed via the TAA control group. Therefore, hepatic cells were well-conserved, approximately usual liver lobular construction by central veins besides burning liver strings. Outcomes confirmed the protective roles of Silymarin in contrast to the TAA-induced liver impairment. Microscopically inspection of liver sections from rats fed with Alpinetin discovered diminished grades of fibrosis; however, a lesser necrotic tissue facilities parenchyma, considerable stages of typical parenchymal construction with tinny fibrous, and minor amounts of cytoplasmic vacuolization and nucleic damage.

### 3.7. Immunohistochemically Staining of Hepatic Slices

TAA-induced liver injury plus significance Alpinetin examined via immunohistochemically stain PCNA, and α-SMA appearance liver parenchyma utilizing exact antibodies. Usual control group existing down-regulation of PCNA and α-SMA stain, demonstrating no cell regeneration fashionable (Figures 5 and 6). In contrast, the TAA control group had remarkable PCNA and α-SMA expression.

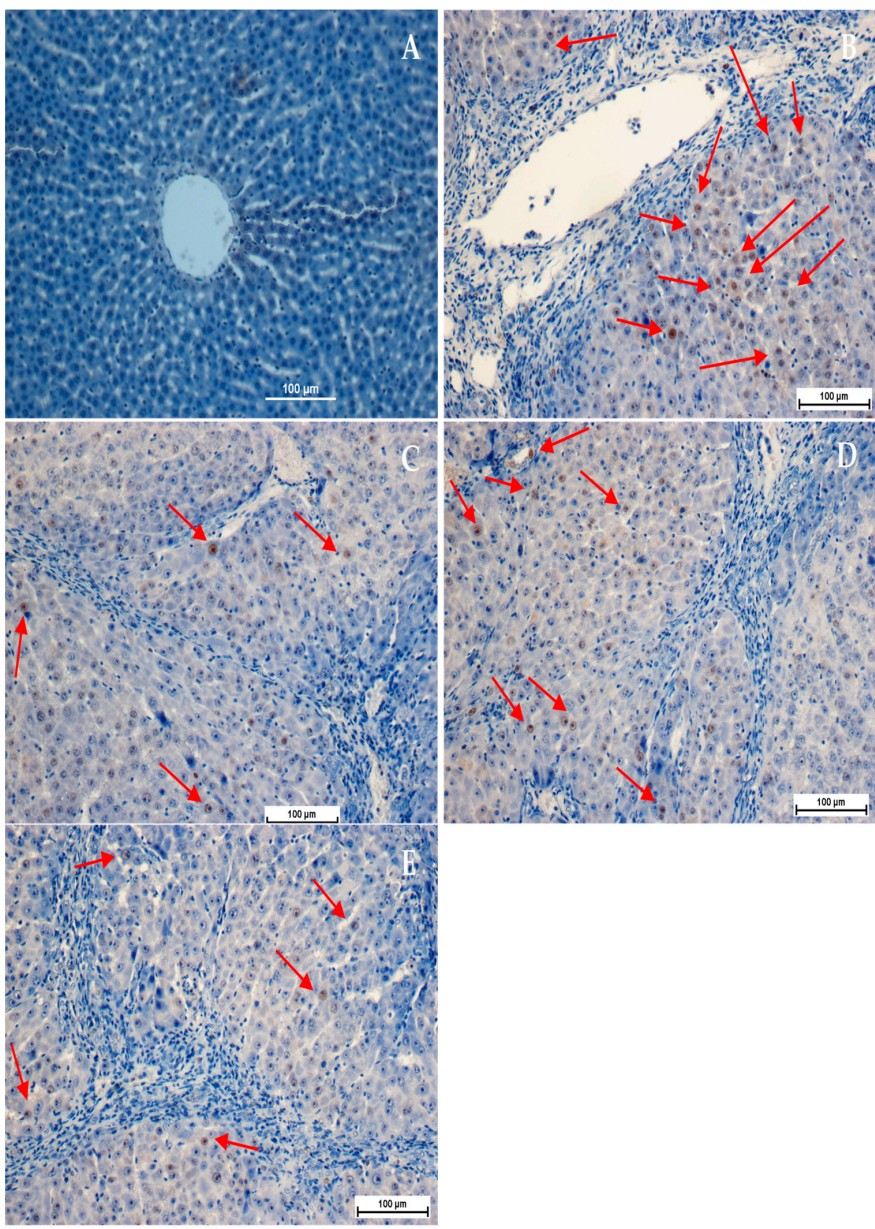

**Figure 5.** Influence of Alpinetin on PCNA stains. (**A**) Normal control group, no PCNA expression;

(**B**) TAA-fed group showed severe fibrosis with greater PCNA expression in the necrotized hepatocytes appearance hepatic tissue (**C**) TAA + Silymarin group, minor PCNA-stained hepatocytes (arrow) indicating less hepatocyte proliferation; (**D**) TAA + 30 mg/kg Alpinetin, insignificant PCNA appearance as indicated by moderate PCNA staining (arrow) in the hepatocytes; (**E**) TAA + 60 mg/kg Alpinetin, minor PCNA expression (arrow) with few proliferated necrotized hepatocytes which were observed in the liver (PCNA stain 40×).

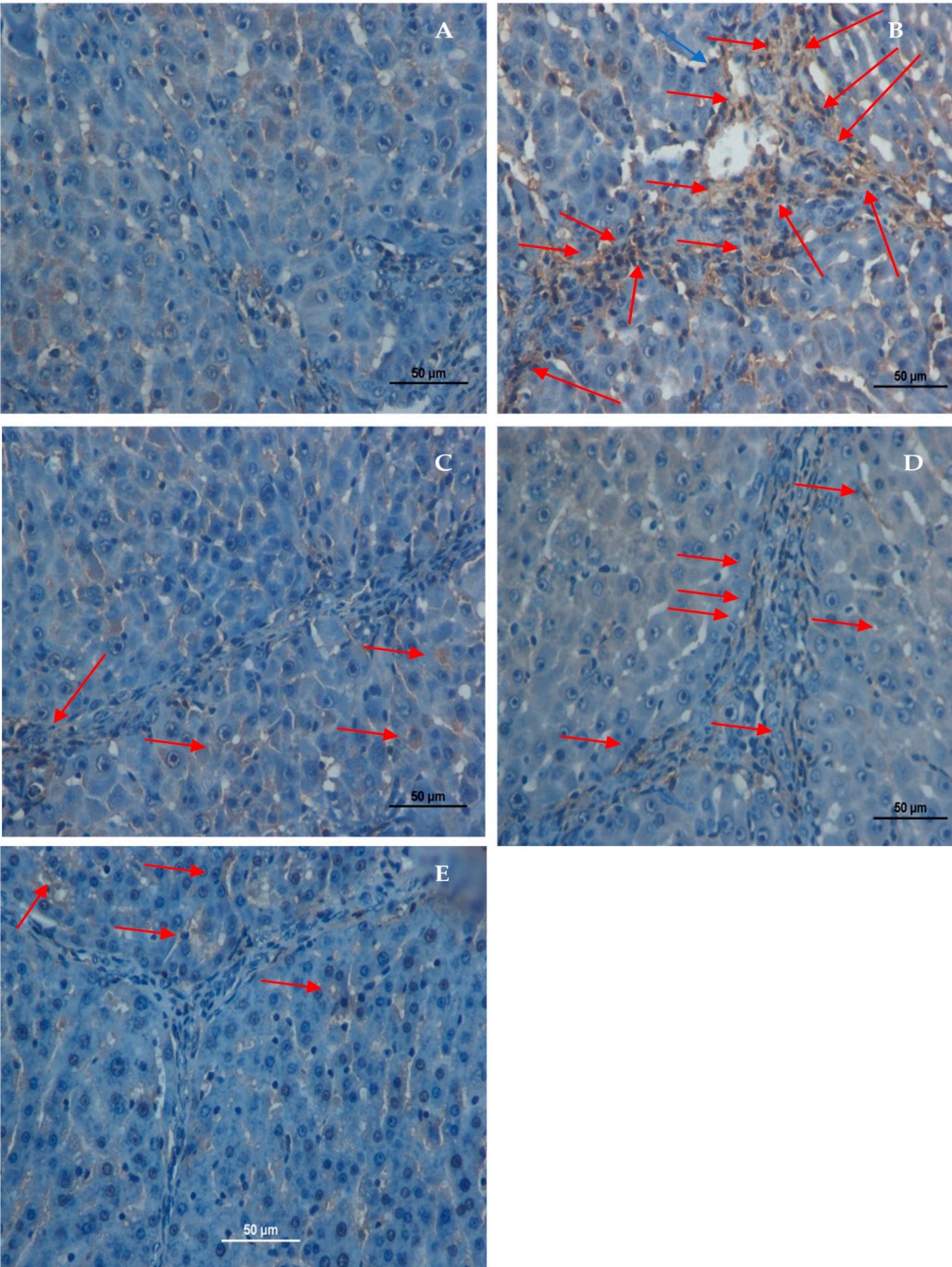

**Figure 6.** Effect of Alpinetin on α-SMA stains. (**A**) Normal control group, no α-SMA appearance; (**B**) TAA-fed group, overexpressed α-SMA staining (arrow) in hepatic tissue (**C**) TAA + Silymarin group, minor α-SMA staining (arrow) appearance; (**D**) TAA + 30 mg/kg, Alpinetin, insignificant α-SMA staining (arrow) appearance in hepatic tissue; (**E**) TAA + 60 mg/kg, Alpinetin, minor α-SMA appearance (α-SMA stain 40×).

### 3.7.1. PCNA Stains of Liver Slices

Influence of Alpinetin cell propagation subsequent TAA-induced liver damage detected by immunohistochemical examination PCNA presence liver tissue utilizing anti-PCNA antibody (Figure 5). The hepatic cells normal group presented no PCNA stains, demonstrating no cell regeneration happening. In contrast, the hepatic cells TAA control group up-regulated PCNA countenance besides raised mitotic index, signifying propagation repair of extensive liver tissue injury induced via TAA. Rats gavaged 30 mg/kg Alpinetin, 60 mg/kg Alpinetin, or Silymarin reduced hepatic cell regeneration linked TAA control group, nominated via condensed PCNA appearance a significant declining mitotic index. Alpinetin owed effect PCNA tagging mitotic index dose-dependent scheme.

### 3.7.2. Alpha Smooth Muscle Actin (α-SMA) Stains

α-SMA smooth muscle stains hepatic slices together with trial groups were representative of the stimulated hepatic stellate cells, revealed in Figure 6. α-SMA absent liver normal group (Figure 6A). Extremely stimulated hepatic stellate cells designated via subterranean stains α-SMA liver hepatotoxic control group (Figure 6B). Strength α-SMA stains nearly absent-minded liver Silymarin-fed group (Figure 6C) likened slices, hepatotoxic group. Slow reduction concentration α-SMA stains liver slices in investigational rats fed Alpinetin with very low staining in high doses (Figure 6E), signifying an obvious reduction in the quantity of stimulated hepatic stellate cells.

### 3.8. Influence of Alpinetin on Cytokines Level in Blood

The results of inflammatory cytokine provocative are shown in Figure 7. The TNF-α and IL-6 initiate at extraordinary levels in the TAA control group in contrast to rats fed Silymarin or Alpinetin. Though, rats gavaged with silymarin or Alpinetin expressively condensed levels of TNF-α and IL-6 compared to the TAA control group (Figure 7).

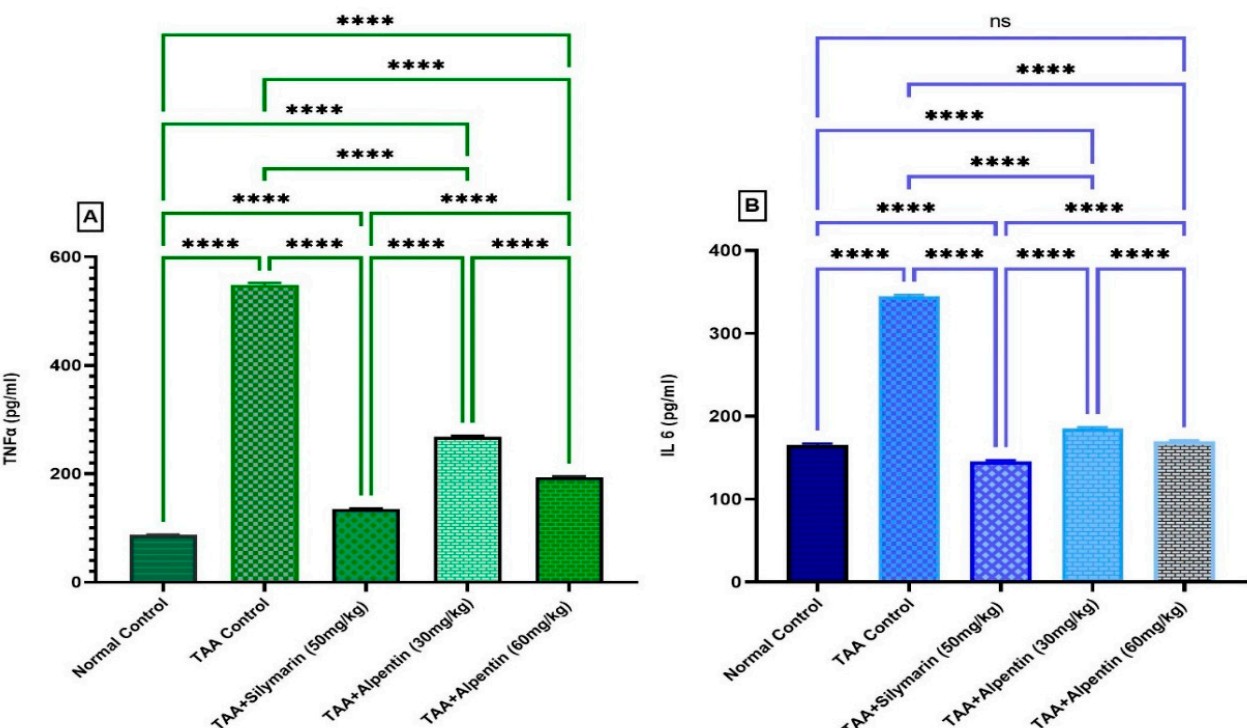

**Figure 7.** Influence of Alpinetin on TNF-α and IL-6 in TAA-induced liver fibrosis. (**A**) TNF-α, and (**B**) IL-6 evaluation. Pre-fed groups: regular control group, TAA control group increased TNF-α, and IL-6, Silymarin group 50 mg/kg, Alpinetin 30 mg/kg, and Alpinetin 60 mg/kg decreased levels TNF-α, and IL-6. Values are obtainable as means (*n* = 6). ns, non-significant; ****, $p < 0.0001$.

## 4. Discussion

The acute toxic study did not display signs of toxicity or death. The study discovered Alpinetin to be safe, besides no toxic gavage by mouth 300 mg/kg observation, with reliable results and additional training utilizing diverse plant extracts or their active ingredients [4,10,30,35,37]. Our results exhibited mean deduction of body mass TAA control group in contrast to the normal group. On the contrary, liver mass was higher dramatically in the TAA control group. Similar outcomes were described by some previous studies [38,39]. Alpinetin-fed rats, however, were able to reduce liver mass by nearly usual morals. The upshot could owe decreased irritation [8]. Comparable marks were earlier recorded by previous studies [4,10].

Rats in the hepatotoxic group presented hepatomegaly. The augmented liver mass/body mass proportion in the TAA control group owed to increased deterioration of the liver. Results were similar to former authors who recorded improved liver mass/body mass relation in hepatotoxic rats [6,9,40]. The drop in liver/body weight gotten Alpinetin activities could be because of a decrease in hyperlipidemia [1].

TAA inoculation-induced liver cirrhosis in rats. The plain impairment was improved by Alpinetin feeding. These explanations are in contrast with a study stated earlier [41]. Our results revealed that Alpinetin might dramatically accelerate the recovery of the liver grievance besides knowingly stopping the consequence of TAA, as described by previous studies [13,40]. The TAA control group was escorted through an obvious upsurge in serum indicators ALP, ALT, AST, total protein, and bilirubin levels. Elevation of serum liver biomarkers reproduced hepatocellular damage. Morals meaningfully reduced roughly to near normal morals upon Alpinetin-gavaged groups. Designates maintenance of plasma–membrane healing hepatic tissue injury persuaded via oral gavage of TAA. Similar clarifications serum liver indicators enhancement via utilizing diverse remedial plants previously knowledgeable [6,10,30]. Hepatoprotective actions perhaps since of its influence contrary cell escape and damage valued goodness hepatocytes sheath. TAA identified suspension RNA initiative from the nucleus to cytoplasm, beginning covering damage which results in an increased declaration of serum liver indicators [37]: current learning, total albumin quantities in serum condensed TAA control group. However, Silymarin or Alpinetin-fed groups return albumin to around normal levels. Similarly, plentiful researchers showed rats' gavaged Silymarin or numerous herbal extracts transported albumin and normal protein levels [2,30,34,42].

It is well recognized that with subsequent cellular impairment, the bulk to create proteins is diminished, and the degree of harm rises, increasing the values of these proteins in the blood [11]. Similar to the results of our study, where total protein then total albumin levels were distinctly reduced, however, as understood Silymarin feeding group, Alpinetin likewise brings backstage protein to normal levels suggesting reappearance from the irregular state to normal. Similarly, numerous investigators showed rats that obtained Silymarin or medicinal plant extracts increased albumin in addition to protein [4,6,15,43,44]. Thioacetamide is identified as hepatotoxic; it produces liver cell death necrosis in higher dosages via the production of free radicals throughout TAA breakdown, resultant oxidative tension arbitrated severe hepatitis, and influences programmed cell death of hepatocytes [13,45]. Hence, it upsurges oxygen-free radicals initiating oxidative pressure besides beginner's programmed cell death; accordingly, raised hepatic enzymes (ALT, AST) indicators cellular hepatic mortification [4,6,14,46].

TAA, as utilized in the current study, and well-recognized for bringing hepatotoxicity in trial rats [47]. The atypical reform lobular construction, the presence of extensive feast fibrosis accumulation, and nodule abrasions of liver tissue were chief appearances of hepatic fibrosis [48]. The liver acts as a foremost character in defensive and purifying the body from external materials [49]. Liver cirrhosis is a chief disease related to numerous pathology procedures counting advanced fibrosis and portal high blood pressure, in addition to carcinoma [50]. Free radical cohort, mitochondria dysfunction diminution anti-oxidants principal development fibrosis [51,52]. Similarly, in outcomes of the present research,

countless scientists described hepatoprotective properties of innumerable plant extracts against TAA-induced hepatic injuries [4,10,15,42].

In the existing study, SOD and CAT activities in hepatic homogenates of Alpinetin-fed rats were suggestively high in TAA groups. Alpinetin meaningfully raised the concentration of CAT also SOD, even though it suggestively declined liver MDA in comparison to the TAA control group. MDA quantity increase in tissue signifies unintended directed lipid peroxidation [40]. Rise of hepatic MDA improved lipid peroxidation leading to decreased anti-oxidant protection activities and avoiding the building of extra free radicals [11,53]. In our finding, TAA tempted an increased TAA control group, and decreased lipid peroxidation was visible by significantly reducing the MDA rate Alpinetin preserved. Similar to the results of existent learning, numerous researchers reported reductions in MDA in rats that received different plant extracts [46]. The diminished liver anti-oxidant enzyme actions in the TAA control group might clarify raised MDA lipid peroxidation.

TAA inoculation formed liver cirrhosis in rats. Nevertheless, rats nourished with Alpinetin might expressively hasten recovery of liver damages evocatively evading the effect of TAA destructiveness. The significance of our research is also similar to previous studies described by numerous co-researchers utilizing different medicinal plants discrepancy TAA-produced liver injury in animals [10,31,37,42]. Outcomes present research demonstrated reduced collagen-building synthesis in Alpinetin-gavaged rats as obtained by Masson trichrome stained hepatic slices. Consequences similar to previous information showed a reduction in collagen fibers contrary to TAA-induced liver fibrosis utilizing curative herbal extracts [15,36,40].

Oxidative pressure and lipid peroxidation show a significant part of beginning cirrhosis in the TAA control group. TAA is absorbed in the body, and its toxic metabolites cause chronic liver damage and the accretion of collagen in ECM, leading to liver fibrosis [30,54].

The detection of proliferating cell nuclear antigen (PCNA) utilizing immunohistochemistry methods is the greatest cooperative means to understand the reproducing action of tissues. PCNA recently documented utilizing polymerase S adding protein [30,42]. In the present research, the normal liver group or silymarin-feeding group showed no important PCNA staining, suggesting the non-appearance of hepatocyte renewal. Over-regulation PCNA presence hepatic cells observed TAA control group, demonstrating widespread proliferation and possible exertions return liver damage [4,55]. Then, rats' gavaged silymarin or Alpinetin displayed expressively condensed cell propagation quantities since reduction PCNA staining.

TAA group, TAA made reactive-oxygen-species (ROS) creating stimulation hepatic satellite cells (HSC) chief basis extracellular matrix construction (ECM) long-lasting hepatic cirrhosis over-regulation of α-SMA. Stimulation HSC escorted propagation besides upgrading ECM building and the presence of α-SMA myofibroblasts [4,10,55,56].

Alpinetin-fed groups showed a down-regulate appearance of α-SMA likened hepatotoxicity group displayed obvious over-regulation of α-SMA. Alpinetin meaningfully inhibits HSC stimulation by evading the production of ROS. Abundant reviews via various researchers recognized down-regulation α-SMA TAA-induced hepatic cirrhosis [4,57,58].

It has been revealed by earlier investigators that TAA-induced liver cirrhosis also induced inflammatory response initiated active chain immune responses related statement huge quantities of inflammatory cytokines, for example, TNF-α and IL-6, to yield the augmented amount of reactive oxygen species [59].

TNF-α, the existence chief pro-inflammatory cytokine formed by macrophages, attracts neutrophils place liver damage [60]. IL-6 is an additional significant pro-inflammatory cytokine revealed to facilitate immune response in addition to acute inflammation. IL-6 stimulates granulocytes besides a granulocyte, successively initiating stress response damaged matter [61,62], in addition, prevents TNF-α construction. Previous studies displayed that TAA is capable of increasing pro-inflammatory cytokines and also reducing anti-inflammatory cytokines in liver tissue [10]. Outcomes of the present study promise explanations; contact with TAA presented raised TNF-α besides IL-6, associated with

the normal control group [15]. Nevertheless, Alpinetin pre-treatment suppressed the promotion of TNF-$\alpha$ and IL-6 amounts, demonstrating the anti-inflammatory influence of TAA-induced liver cirrhosis rats [4]. Similar to previous histopathological outcomes, fewer inflammatory responses were detected in Alpinetin-fed rats through liver cirrhosis. As formerly specified, IL-6 excites neutrophils, monocytes, and lymphocytes at the site of inflammation [15]. IL-6 induced the manufacture of the greatest acute phase proteins in inflammatory reactions. Alpinetin uses a hepatoprotective consequence via an anti-inflammatory mechanism.

The mechanism of action of Alpinetin might be through free radical hunting and slaking establishment of single oxygen, which defends the liver against oxidative tension and motivates liver healing and anti-inflammatory mechanisms.

## 5. Conclusions

The present research displayed that Alpinetin has hepatoprotective effects in TAA-induced liver fibrosis in rats established via gross morphology, histopathology, biochemical parameters, increased endogenous enzymes, SOD and CAT, decreased oxidative stress and lipid peroxidation (MDA), down-regulation of PCNA and $\alpha$-SMA, modulation of inflammatory cytokines, and own anti-oxidant and free radical scavenger properties.

**Author Contributions:** Conceptualization, M.A.A. and S.H.S.; methodology, M.A.A., S.H.S. and A.R.A.; software, S.H.S.; validation, M.A.A., S.H.S. and I.A.A.I.; formal analysis, S.H.S.; investigation, M.A.A., S.H.S. and D.N.F.A.; resources, S.H.S. and M.A.A.; data curation, S.H.S. and A.S.M.J.; writing—original draft preparation, M.A.A. and S.H.S.; writing—review and editing, S.H.S., M.A.A., A.R.A., I.A.A.I., A.S.M.J. and D.N.F.A.; visualization, S.H.S. and A.S.M.J.; supervision, S.H.S. and M.A.A.; project administration, S.H.S. and M.A.A.; funding acquisition, I.A.A.I.; All authors have read and agreed to the published version of the manuscript.

**Funding:** This research received no external funding.

**Institutional Review Board Statement:** Not applicable.

**Informed Consent Statement:** Not applicable.

**Data Availability Statement:** The data used to support the findings of this study are included within the article, and further information is available from the corresponding author upon request.

**Conflicts of Interest:** The authors declare that there is no conflict of interest regarding the publication of this paper.

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
