# Peer review of "Hepatoprotective Effect of Alpinetin on Thioacetamide-Induced Liver Fibrosis in Sprague Dawley Rat"

_applsci, doi:10.3390/app13095243_

Round 1
Reviewer 1 Report
The manuscript on Hepatoprotective Effect of Alpinetin on Thioacetamide-Induced Liver Fibrosis in Rat is quiet interesting role of alpinetin ameliorates TAA induce liver fibrosis in rodent model.
However, there are few quires authors need to addresses as follows
In abstract - toxicity study should be included
In introduction section - source and biological importance of alpinetin should be included
In material and methods - Rodent model strain name should be included. Nowhere mentioned in the manuscript
Results - Images are not clear and visible. Results should be discussed in detail
Overall - English language to be improved and some sentences are meaningless
Author Response
Dear Reviewer
The manuscript corrected, thanks for your comments.

Reviewer 2 Report
The research article by Suhayla Hamad Shareef et al., presented the ‘Hepatoprotective Effect of Alpinetin on Thioacetamide-Induced Liver Fibrosis in Rat. Authors have displayed the hepatoprotective role of Alpinetin and showed the drug can improve defensive mechanisms against hepatic tissue injury. However, the current version of the manuscript has lot of shortcomings. A high degree of plagiarism was also observed (32%). Since the manuscript has good data, I would encourage resubmission after following the recommendations.
Major recommendations:
The abstract may be restructured by a mention of a brief outline of results may enhance the first-sight quality of the research article.
The language used throughout the manuscripts very poor and needs a major revision to conform to standard technical English language.
The language used in the experimental set-up and data presentation is not easily understood
The discussion section may be further refined to present a clear, coherent defence of your findings.
The discussion part is very poor. Authors need to correlate their results with recent publications in a similar types of research to highlight the novelty of work
Authors need a careful and scientific ways to correlate their data with existing data and also
carefully rectify grammatical mistakes
All the figures need to be properly labelled to show the results. Presentation of results in
figures are poorly represented
Legends in figure 5 are not mentioned
The quality of figures needs to be enhanced
References need minor corrections and modifications as per journal guidelines
Minor recommendations:
Page 2:
Line 51-52 paraphrase to a meaningful sentence
TAA is hepatotoxic, the single dose is accepted to be able to yield centrilobular
hepatocyte necrosis and continuing government led to liver fibrosis in laboratory animals [7, 8].
Lines 55 to 56: paraphrase a meaningful sentence
Interestingly, an extensive examination had been hurled to classify novel hepatoprotective remedies from ordinary bases.
Lines 56 to 57: paraphrase a meaningful sentence
Aromatic plants and therapeutic basic medication materials pondered possible basis battle
several illnesses, together with liver cirrhosis.
Line 61: Paraphrase what authors try to mention in this sentence
“ingredient through helpfulness anticancer effects”
Line 61: Is this finding simply an assumption
“The present study is assumed to evaluate conceivable hepatoprotective possessions of
Alpine in TAA-induced liver cirrhosis in rats”
Line 70: I hope the following number is correct; please confirm ?
Ethic number ERB, 115, 11/03/2021
Author Response
Dear Reviewer, The manuscript corrected, thanks for your recommendations.

Reviewer 3 Report
The article by Shareef and colleagues entitled “Hepatoprotective Effect of Alpinetin on Thioacetamide-Induced Liver Fibrosis in Rat”. They have provided helpful information about Alpinetin as a protective agent against Thioacetamide-Induced Liver Fibrosis.
I have read the manuscript thoroughly, their findings look exciting and informative, and they have performed various techniques to validate their hypothesis. Nevertheless, the present manuscript needs to be improved before acceptance.
Comments or suggestions
· The authors have numerous grammatical, spelling, and inconsistencies throughout the manuscript that must be adequately proofread.
· I will suggest rewriting the conclusion in the abstract section correctly and concisely in an abstract.
· I will suggest adding more about fibrosis in the introduction section.
· Number of animals used is missing in table1-4
· In figure 1, the authors need to add a scale bar.
· The abbreviation used to indicate significance (a,b) should be described
· In figure 3 & 4 (C, D, and E images are missing
· In figure IHC, I will suggest adding slides with better resolution.
· I will suggest adding a graphical abstract in the manuscript which can be the point of attraction to the readers.
Author Response
Dear Reviewer, thanks for your comments I added and corrected all things you mentioned in your comments.

Reviewer 4 Report
There are a few minor English discrepancies - for example the fibrous tissue propagation described that might suggest a tumor-like behaviour, which is not the case.
Is it really necessary to describe the preparation of reagents divided in separate paragraphs?
Which specific type of rats was used?
Why were the rat groups assigned letters and after they were again presented as "vehicle" or "Alpinentin"?
The aspect of tables should be uniform (1-4 vs 5-7 are different). Also, the superscript symbols of "a" and "b" are not defined anywhere.
Figures 3 and 4 are incomplete.
Mechanism action of Alpinentin is speculative.
Author Response
Dear Reviewer, The manuscript corrected, thanks for your suggestions
